

# *ZmIAA5* regulates maize root growth and development by interacting with *ZmARF5* under the specific binding of *ZmTCP15/16/17*

Feiyang Yang[1,*], Yutian Shi[2,*], Manli Zhao[2], Beijiu Cheng[2] and Xiaoyu Li[2]

[1] College of Agronomy, Anhui Agricultural University, Hefei, Anhui, China
[2] School of Life Sciences, Anhui Agricultural University, Hefei, Anhui, China
[*] These authors contributed equally to this work.

Corresponding authors
Beijiu Cheng, cbj@ahau.edu.cn
Xiaoyu Li, lixiaoyu@ahau.edu.cn

## ABSTRACT

**Background**. The auxin indole-3-acetic acid (IAA) is a type of endogenous plant hormone with a low concentration in plants, but it plays an important role in their growth and development. The *AUX/IAA* gene family was found to be an early sensitive auxin gene with a complicated way of regulating growth and development in plants. The regulation of root growth and development by *AUX/IAA* family genes has been reported in Arabidopsis, rice and maize.

**Results**. In this study, subcellular localization indicated that ZmIAA1-ZmIAA6 primarily played a role in the nucleus. A thermogram analysis showed that *AUX/IAA* genes were highly expressed in the roots, which was also confirmed by the maize tissue expression patterns. In maize overexpressing *ZmIAA5*, the length of the main root, the number of lateral roots, and the stalk height at the seedling stage were significantly increased compared with those of the wild type, while the EMS mutant *zmiaa5* was significantly reduced. The total number of roots and the dry weight of maize overexpressing *ZmIAA5* at the mature stage were also significantly increased compared with those of the wild type, while those of the mutant *zmiaa5* was significantly reduced. Yeast one-hybrid experiments showed that *ZmTCP15/16/17* could specifically bind to the *ZmIAA5* promoter region. Bimolecular fluorescence complementation and yeast two-hybridization indicated an interaction between ZmIAA5 and ZmARF5.

**Conclusions**. Taken together, the results of this study indicate that *ZmIAA5* regulates maize root growth and development by interacting with *ZmARF5* under the specific binding of *ZmTCP15/16/17*.

## INTRODUCTION

Cereals provide essential dietary energy and nutrients for humans and animals (*Jiang et al., 2021*). The yield and quality of cereal grains are dependent on the coordinated regulation of endosperm development, the uptake of photosynthate and the production of storage molecules (*Yamauchi et al., 2019*). Maize (*Zea mays* L.) is one of the three major crops in the world, providing food for more than 4.5 billion people in 94 developing countries, so

it plays a very important role in international food security (*Sanchez, Rasmussen & Porter, 2014*). Additionally, maize is widely used in agriculture, animal husbandry and industry, for items such as food, animal feed and biofuel production (*Shiferaw et al., 2011*). Therefore, maize has great economic value, and research on the molecular mechanism of maize growth and development will help producers better guide its daily production (*Basunia et al., 2021*).

Auxin/indole-3-acetic acid (IAA) is one of the most studied plant hormones among many plant hormones, and scientists have been studying auxin for more than 70 years (*Kasahara, 2016*). Auxin was one of the earliest plant hormones discovered by humans, and it plays an important role in many aspects of plant growth and development as well as in various biological processes. At present, more in-depth studies include plant apical dominance, cell division, vascular tissue formation and root hair development (*Ciarkowska, Ostrowski & Jakubowska, 2018*; *Brumos, Alonso & Stepanova, 2014*). Several mechanisms for maintaining auxin homeostasis have been identified in plants (*Ludwig-Muller, 2011*). Multiple studies have shown that *AUX/IAA* proteins and auxin-responsive factor (ARF)-mediated signaling play an important role in regulating plant root formation (*Rouse et al., 1998*). ZmIAA10 can interact with ZmARF25 and ZmARF34, thereby affecting root growth and development in maize (*Deng et al., 2014*). This is a well-studied *AUX/IAA* family gene that regulates root growth and development in maize.

The root system is one of the most important organs in plants. The root system has the function of absorbing water and nutrients from the soil, and its growth state affects its ability to perform its functions (*Deng et al., 2014*; *MacDonald et al., 2011*). Maize root development is adjusted by the integration of endogenous factors, such as phytohormones, with integrated environmental stimuli, such as soil nutrients (*Hermans et al., 2006*). In previous studies, scientists identified some key genes involved in maize root growth and development by screening maize root mutants (*Hill et al., 2006*). Among them, the genes involved in the establishment of the top-basal longitudinal pattern of the radicle include MP, HBT, BDL, etc. (*Lynch, 2007*; *Vance, Uhde-Stone & Allan, 2003*). Auxin synthesis, transport and signal transduction have also been shown to be involved in regulating maize root growth and development (*Nestler, Keyes & Wissuwa, 2016*). For example, the rtcs mutant was the first maize root mutant discovered by scientists, and the *RTCS* gene was found to be related to the auxin signal transduction pathway (*Stetter, Schmid & Ludewig, 2015*). In addition, some researchers indicated that auxin plays a key regulatory role in the formation of maize lateral root primordia and the early stage of lateral root growth (*Fan et al., 2003*; *Ramaekers et al., 2010*).

In this study, we performed a phenotypic analysis of *ZmIAA5* gene overexpression and mutant plants. Through promoter binding experiments and identification experiments on interacting proteins, we found that *ZmIAA5* regulates maize root growth and development by interacting with *ZmARF5* under the specific binding of *ZmTCP15/16/17*.
## METHODS AND MATERIALS

### Database analysis

The *AUX/IAA* homologs in maize protein sequences of Arabidopsis and maize (Zea mays L) were obtained from the following databases: Phytozome (https://phytozome-next.jgi.doe.gov/), Uniprot (https://www.uniprot.org/), and Tair (https://www.arabidopsis.org/) (*Wang et al., 2015*). The Arabidopsis protein sequence was used as the initial query sequence, and the databases were searched with MultAlin (http://multalin.toulouse.inra.fr/multalin/). The phylogenetic tree was constructed by using the neighbor-joining method in MEGA7 based on the amino acid sequences. Six orthologous Arabidopsis AUX/IAA proteins in maize were characterized and named *ZmIAA1, ZmIAA2, ZmIAA3, ZmIAA4, ZmIAA5* and *ZmIAA6*.

### Transcriptome analysis

We processed the data as described in a previous article that was published in the International Journal of Molecular Science (*Wang et al., 2015*); specifically, we downloaded transcriptome data from Plexdb (ZM37) and used Nimble Gen chip technology to generate a genome-wide gene expression map of the maize inbred line B73 to analyze the spatiotemporal expression patterns in *ZmIAA1-ZmIAA6* genes during development (*Wang et al., 2015*). The microarray data for the six genes *ZmIAA1-ZmIAA6* were imported into Bioconductor and R for expression analyses, and then the pheatmap package was used to make the heat maps (*Wang et al., 2015*).

### Plant growth and phosphorus treatments

Maize ecotype B73 and KN5585 were as an important experimental material used in this study. We performed experiments as described previously in a paper published in Plant Science (*Kong et al., 2019*), specifically maize seedlings were grown in Hoagland's liquid medium for 2 weeks at 30 °C on a 14-hour light/10-hour dark cycle (*Kong et al., 2019*). The formula of Hoagland liquid medium mainly includes macro elements and trace elements. Macro elements mainly includes Calcium nitrate tetrahydrate (945 mg/L), Potassium nitrate (607 mg/L), Ammonium dihydrogen phosphate (115 mg/L), Magnesium sulfate heptahydrate (493 mg/L); Trace elements mainly includes Ethylenediaminetetraacetic acid disodium iron (20–40 mg/L), Boric acid (2.86 mg/L), Manganese sulfate tetrahydrate (2.13 mg/L), Zinc sulfate heptahydrate (0.22 mg/L), Copper sulfate pentahydrate (0.08 mg/L), Ammonium molybdate tetrahydrate (0.02 mg/L). Roots, stems, leaves, filament, cluster, embryo, and bract of maize were used for tissue-specific expression analysis. The Arabidopsis ecotype Col-0 was used as an important experimental material In this study. Arabidopsis seeds should first be sterilized in a solution containing 12% sodium hypochlorite for 10 min, then washed 6 times with sterile water, and finally sown in 0.8% Agar on 1/2 Murashige and Skoog (MS) medium (*Kong et al., 2019*). The medium was placed at 4 °C for 3 days, then grown vertically under standard conditions (16 h light/8 h dark cycle at 22 °C) for 7 days, and finally the Arabidopsis seedlings were photographed and data Measurement (*Kong et al., 2019*).

## Phenotypic and statistical analyses

The root phenotypes were observed from maize seedlings cultured in liquid medium for 14 days and mature maize grown in the field for 65 days. After observing the root phenotype of maize cultivated in liquid medium for 14 days, we transplanted the maize seedlings to a natural environment, cultured them for 65 days, and then removed them from the soil to observe their root phenotypes. The statistical analysis of various phenotypes requires a unique correlation analysis. The data correlation test was performed using the GraphPad PRISM computer program Form 7.0 (GraphPad software) with reference to previous research methods (*Wang et al., 2015*). One-way or two-way ANOVA made accurate with Tukey's multiple contrast test was used to determine the statistical importance.

## Subcellular localization analysis

We performed the experiments as described previously in a paper published in Plant Science (*Kong et al., 2019*). We used a Zeiss LSM 710 META laser scanning microscope owned by our group to detect the fluorescence in the root apex and we adjusted the wavelength of the confocal microscope from 610 to 630 nm to display the propidium iodide signal (*Kong et al., 2019*). Prior to observation, the root tip was treated with 20% sucrose for 20 min to induce plasmolysis of the root tip cells, and the GFP was observed over the wavelength range of 450 to 470 nm. A fluorescence analysis was performed using ZEN 2009 software as described above (*Kong et al., 2019*).

## RNA extraction and qRT-PCR analysis

After collection, we performed the experiments as described previously in a paper published in Plant Science (*Kong et al., 2019*). Specifically, samples should first be immediately frozen in liquid nitrogen and stored in a $-80\,°C$ freezer for RNA extraction (*Kong et al., 2019*). We extracted the total RNA from the samples using the RNAprep Pure Plant Kit (Tiangen) (*Kong et al., 2019*). The experimental steps were performed in sequence according to the manufacturer's instructions (*Kong et al., 2019*). Quantitative real-time RT-PCR-related experiments were performed using a Lightcycler 480 SYBR Green I Master Mix (Roche) and an Applied Biosystems 7300 Real-Time PCR System. ZmActin1 and AtActin2 were used as internal controls. Note that to ensure the reliability of the experiment, we set up three biological replicates for each sample in the experiment (*Kong et al., 2019*).

## Yeast one-hybrid assays

We performed yeast single-hybrid experiments with reference to a study published in PPRC (*Zhang et al., 2022*). Throughout the experiments, we used the pAbAi + pGADT7-Rec/p53 combination as a positive control (*Zhang et al., 2022*). First, the promoter fragment of the *ZmIAA5* gene was cloned into the pAbAi vector, and second, the full-length CDS of *ZmTCP15/16/17* was cloned into the pGADT7 vector (*Zhang et al., 2022*). Subsequently, the resulting recombinant plasmid was transformed into yeast (Y1H) cells (*Zhang et al., 2022*). The yeast (Y1H) was spread on nonselective SD-Leu-AbA solid medium and placed in an incubator at 30 °C for 3–4 days, and then the yeast (Y1H) was transferred onto selective SD-Leu + AbA (900 ng/mL) solid medium and grown for 4–6 days. Note that the

initial concentration of yeast was adjusted to 1 at an OD of 600 and then diluted to 1/10, 1/100, and 1/1,000 (*Zhang et al., 2022*).

## Yeast two-hybrid assays

We performed the experiments as described previously in a paper published in Plant Science (*Kong et al., 2019*). Specifically, we first needed to construct the *ZmARF5/7/25*-pGADT7 vector and the *ZmIAA5*-pGBKT7 vector. The recombinant plasmids of the above genes were transformed into yeast (Y1H) cells, and the specific steps were similar to those used in the yeast one-hybrid experiments (*Kong et al., 2019*). The yeast (Y1H) cells were placed in an incubator, and when a single colony visible to the naked eye grew on the medium, a single colony was picked, and the colony was spotted onto SD/-Trp-Leu-His-Ade-deficient solid medium (*Kong et al., 2019*). The culture plate was placed in an incubator to continue culturing, and its plaque growth was observed (*Kong et al., 2019*). Note that the concentrations must be set up at 1/10, 1/100, and 1/1,000 according to the concentration ratio of 10:1 (*Kong et al., 2019*).

## Bimolecular fluorescence complementation assay

Double digestions of pUC-SPYCE and pUC-SPYNE were performed with the restriction enzymes BamHI and SalI. *ZmIAA5*-pUC-SPYCE and *ZmARF5/7/25*-pUC-SPYNE vectors were constructed, recombined and ligated and transformed into maize protoplasts (*Coelho et al., 2010*). The mixed plasmid system was placed in the dark for induction transformation, and the induction transformation time was approximately 48 h. Lastly, we used laser confocal microscopy to observe whether the cells produced interaction signals (*Kamran et al., 2018*).

## Statistical analysis

This article also involved a series of statistical analyses. The biostatistical analysis was primarily performed on the root traits of the overexpressed and mutant maize, including the length of the primary root, length of the seminal root, number of primary lateral roots, number of total roots, heights of 14-day-old plants and weights of 65-day-old plants. We performed three biological statistics in total, each with a sample size of 10 pairs of the root traits listed above. The statistics are accurate to three decimal places where possible. We compared the overexpressed strains and mutants with the wild type and obtained the final difference results, and we performed hypothesis tests on the results and calculated the $p$ values (*Fageria, 2005*). Statistically, $P < 0.05$ is generally considered to indicate a significant difference, $P < 0.01$ is considered to indicate a significant difference, and $P < 0.001$ is considered to indicate an extremely significant difference (*Ma et al., 2018*).

## RESULTS

### Phylogenetic analysis of identified *Aux/IAA* transcription factors in maize

The existing studies have shown that the *Aux/IAA* gene family affects all aspects of plant growth and development, and recent studies on this family of genes have been reported in Arabidopsis, rice and maize (*Luo, Zhou & Zhang, 2018*). Dynamic changes in the auxin
levels can accurately and rapidly trigger the expression of related response genes, and *Aux/IAA* family genes are this type of response genes (*Goldental-Cohen et al., 2017*). Members of the *Aux/IAA* family have been shown to edit a class of short-lived proteins, primarily in the nucleus. They play a key role in suppressing the expression levels of genes activated or repressed by *ARF* (*Abel, Oeller & Theologis, 1994*). It has also been demonstrated that auxin-mediated transcriptional regulation is largely dependent on the function of *Aux/IAA*.

To explain the functions of IAA/AUX family proteins more systematically, we performed a phylogenetic analysis of 40 transcription factors that encode auxin proteins from maize and 32 from Arabidopsis according to the plant transcription factor databases PlantTFDB and UniProt (*Jiang et al., 2021*). We selected the protein sequences of the *Aux/IAA* transcription factor family and compared them using ClustalW. The evolutionary history was inferred using the neighbor-joining method (*Lavy & Estelle, 2016*; *Jiang et al., 2021*). The optimal tree with a sum of branch length = 9.99286727 is shown. All the positions containing gaps and missing data were eliminated (*Lavy & Estelle, 2016*; *Yamauchi et al., 2019*). Evolutionary analyses were conducted in MEGA7. Aux/IAA transcription factors could be clearly divided into 4 groups according to sequence similarity (Fig. 1A).

We found that Zm00001d021279 (named *ZmIAA1*), Zm00001d041418 (named *ZmIAA2*), Zm00001d013707 (named *ZmIAA3*), Zm00001d034463 (named *ZmIAA4*), Zm00001d030993 (named *ZmIAA5*) and Zm00001d013071 (named *ZmIAA6*) were in the same group as *AtIAA1, AtIAA2, AtIAA3, AtIAA4, AtIAA8, AtIAA9* and *AtIAA27* Arabidopsis transcription factors, indicating that those proteins may have similar functions.

## ZmIAA1-ZmIAA6 proteins share conserved domains

Domains are distinct units of protein molecular evolution and are often associated with proteins that perform specific functions, such as binding or catalysis (*Saitou & Nei, 1987*). Over the process of biological evolution, a single amino acid may change frequently, but it is relatively conserved in terms of the entire protein domain and will not easily change during evolution. Normally, maize Aux/IAA proteins are associated with auxin-responsive factors (ARFs), so *Aux/IAA* genes inhibit the activity of auxin-responsive factors (ARFs) (*Marchler-Bauer et al., 2005*; *Hou et al., 2020*). When the level of auxin in maize increases, the AUX/IAA protein is degraded by ubiquitination, and then *ARFS* is released to activate or inhibit the expression of downstream genes (*Weijers et al., 2005*).

Generally, most Aux/IAA proteins comprise four conserved domains that are responsible for different functional properties, and they are called domains I–IV (*Kalantzopoulos et al., 2018*). Domain I harbors the "LxLxL" motif, domain II contains the "GWPPV" motif and a nuclear localization signal (NLS) (*Gray et al., 2001*), and domains III and IV with the "GDVP" motif and aSV40-type NLS serve as binding regions for homo and heterodimerization with other Aux/IAA proteins and/or ARFs (*Mironova et al., 2013*). We conducted a homology analysis using the MultAlin online tool (http://multalin.toulouse.inra.fr/multalin/) (*Mironova et al., 2013*). Our results show that some amino acids of ZmIAA1-ZmIAA6 are totally identical (Fig. 2A). In addition, we found that these six ZmIAA proteins also share four conserved domains, as reported previously

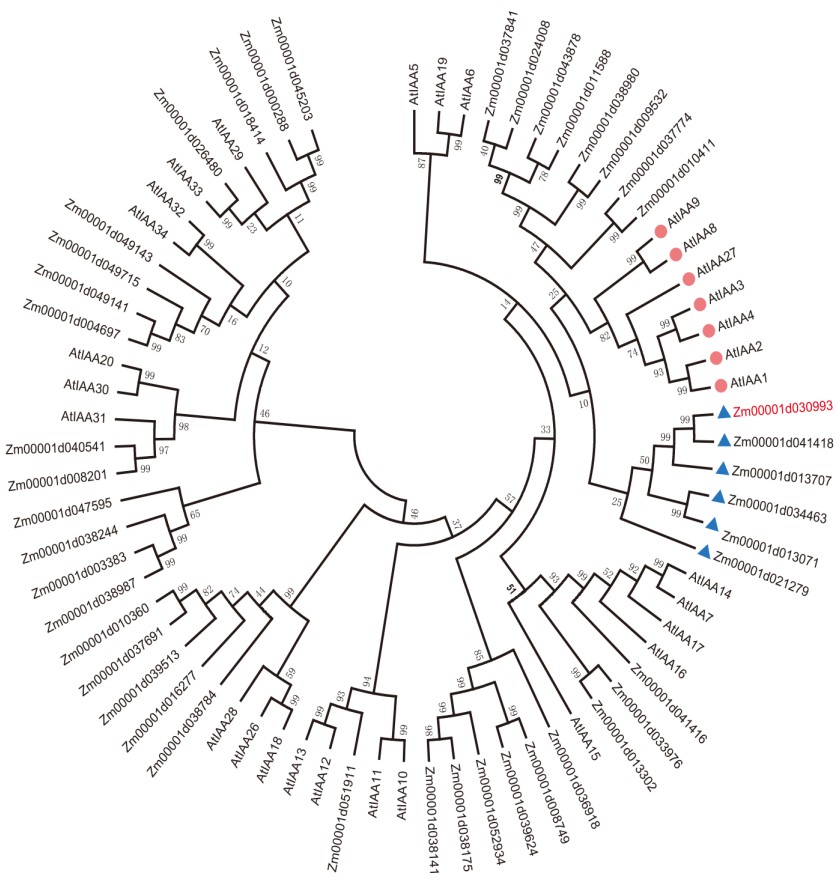

**Figure 1** **Evolutionary relationships of AUX/IAA proteins.** The IAA proteins of maize and Arabidopsis were aligned with ClustalW, and the phylogenetic tree was constructed by using the neighbor-joining method in MEGA9.0. The red diamond represents AUX/IAA protein in Arabidopsis, and the AUX/IAA proteins in six corns surrounding the blue triangle were used as our study subjects. The *ZmIAA5* gene is indicated in red font. All the protein sequences are from UniProt (https://www.UniProt.org/).

(*Kalantzopoulos et al., 2018*; *Gray et al., 2001*; *Mironova et al., 2013*) (Fig. 2B). That is, it is possible that they are also conserved in terms of biological function.

## ZmIAA5 proteins are located in the nucleus and are expressed primarily in roots

The online tool Plant-mPLoc (http://www.csbio.sjtu.edu.cn/bioinf/plant-multi/) was used to predict the subcellular localization of the Aux/IAA transcription factor family proteins located in the nucleus (*Chou & Shen, 2010*). The subcellular localization of ZmIAA1-ZmIAA6 proteins fused to green fluorescent protein (GFP) was identified in a transient assay in maize protoplasts by means of Agrobacterium infiltration. While all the GFP-tagged genes produced fluorescence in the nucleus (Fig. 3A), the experimental results showed that the six genes ZmIAA1-ZmIAA6 were localized in the nucleus.

To verify the accuracy of the heat map data we obtained, we examined the expression patterns of *ZmIAA1-ZmIAA6* in different tissues during maize seed development (Fig. 3B).

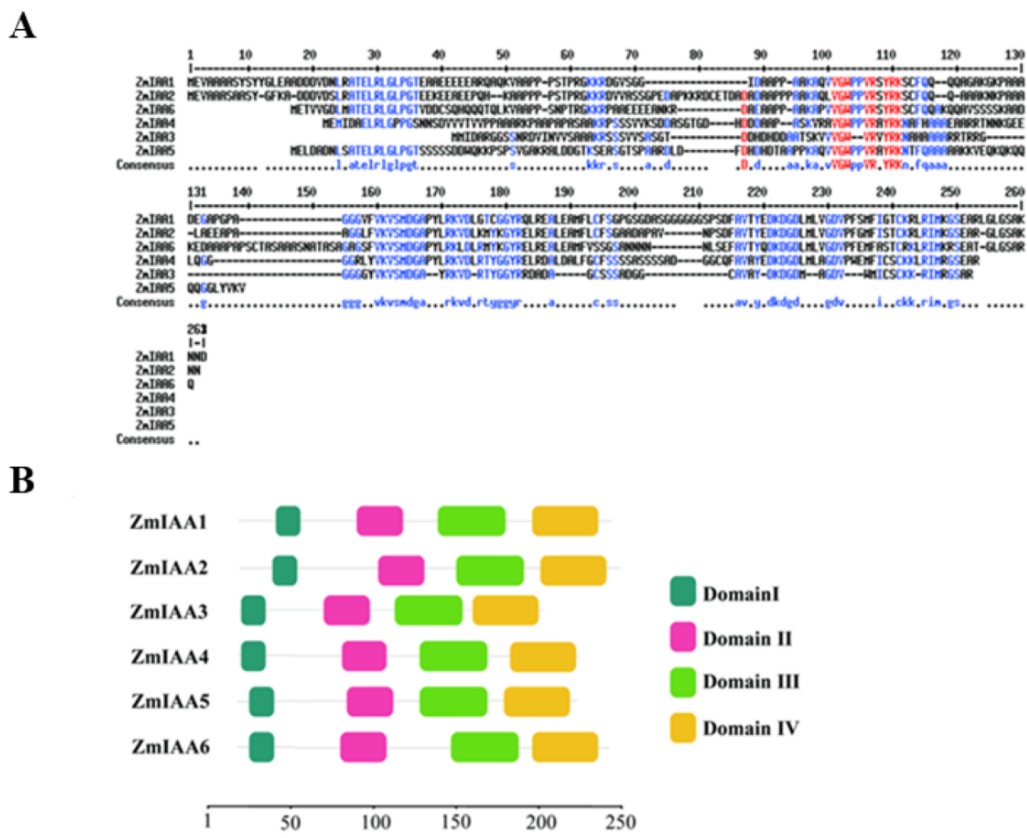

**Figure 2** **AUX/IAA proteins in maize share similar amino acid sequences.** (A) The AUX/IAA protein sequences of maize were aligned with the MultiAlin online tool (http://multalin.toulouse.inra.fr/multalin/). The blue color represents the consensus. (B) Sequence of ZmIAA1-6 proteins. The red color represents the conserved sequences. All the protein sequences are from UniProt (https://www.uniprot.org/). Structure diagram of AUX/IAA proteins. The number represents the amino acid sequence position of the domain.

The experimental results showed that the expression levels of *ZmIAA2*, *ZmIAA5* and *ZmIAA6* in roots were higher and were highly expressed in immature leaves (Fig. 3B). Bar chart statistics also illustrate this point (Fig. 3C). The above experimental results showed that the expression patterns of these genes were consistent with some previous reports.

## Maize mutant *zmiaa5* affects root growth and development

Maize genes can be mutagenized using methanesulfonic acid (EMS) chemical mutagenesis techniques to obtain mutant maize material (*Svitashev et al., 2016*). To investigate whether the *AUX/IAA* gene function is conserved in maize, we obtained two lines of a mutant of *IAA5* in maize by ordering from the website (http://elabcaas.cn/memd/public/index.html#/). These lines were named *zmiaa5-1* and *zmiaa5-2*. The mutant was then obtained by chemical mutagenesis of the maize inbred line B73 by EMS. The mutation site of this mutant is located in the second exon sequence of the *ZmIAA5* gene, which is a mutation from the original cytosine C to thymine T (Fig. S2A). After obtaining the mutants, we performed
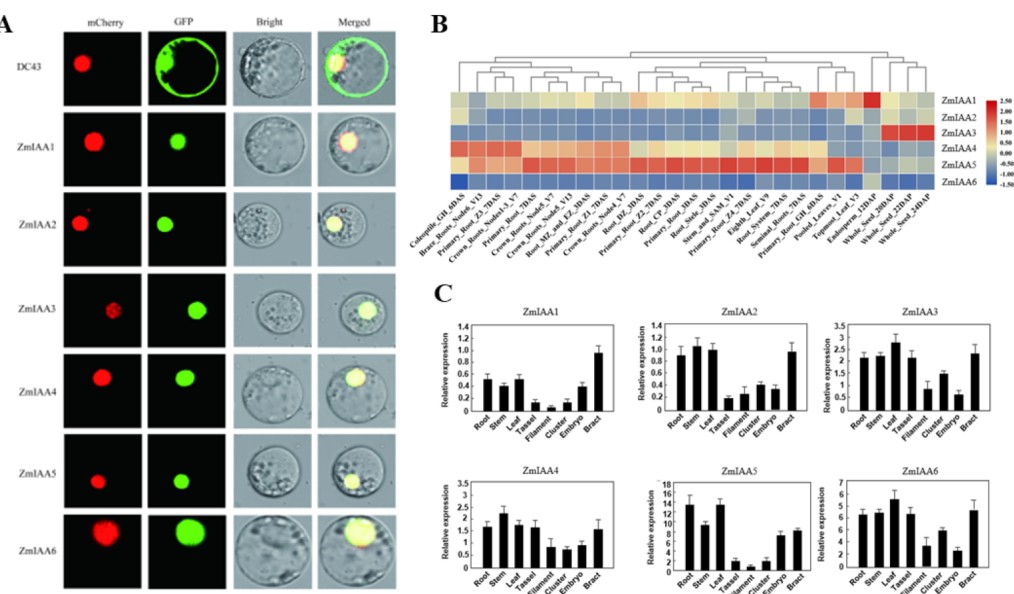

**Figure 3** *AUX/IAA gene expression patterns.* (A) ZmIAA1-ZmIAA6 proteins located in the maize nucleus in the protoplast. The red color represents the mCherry fluorescence signal, the green color represents the GFP fluorescence signal, and the yellow color represents the merged fluorescence signal. (B) The expression of *ZmIAA1-ZmIAA6* in different maize tissues by heat map. Red indicates high expression, yellow represents medium expression, blue indicates very low expression, and numbers represent the signal strength. (C) The expression of *ZmIAA1-ZmIAA6* in different maize tissues was analyzed in the histogram.

two generations of backcrosses to remove the hybrid genetic background and ultimately obtained pure and positive mutant maize (Fig. S2B).

We found that *zmiaa5* mutants exhibited shorter primary roots and fewer lateral roots (Figs. 4A, 4B and 4C). We also performed biological measurements of the taproot length, seed root length, lateral root number, total root number, stem height, and total dry root weight in mature wild-type and mutant maize grown for 14 days in the greenhouse and 65 days in the field. The experimental results indicated that in all the items under the above statistics, the mutant maize data were lower than those of wild-type maize (Figs. 4D–4I). This finding suggests that *ZmIAA5* may play a role in regulating maize root growth and development (*Vanneste & Friml, 2012*).

## Overexpression of ZmIAA5 in maize affects root growth and development

pEC00242 is a medium copy vector with a size of 10.218 kb (*Lee et al., 2017*). The pEC00242 vector primarily contains the following elements: replication elements, bar selection genes driven by the Ubi promoter, DNA sequences of target genes driven by the CaMV35S promoter and Kan antibiotic resistance genes (*Ren et al., 2019*; *Lv et al., 2020*). We used the homologous recombination method to recombine the DNA fragment of the maize *ZmMBL1* gene into the linearized pEC00242 vector to complete the vector construction

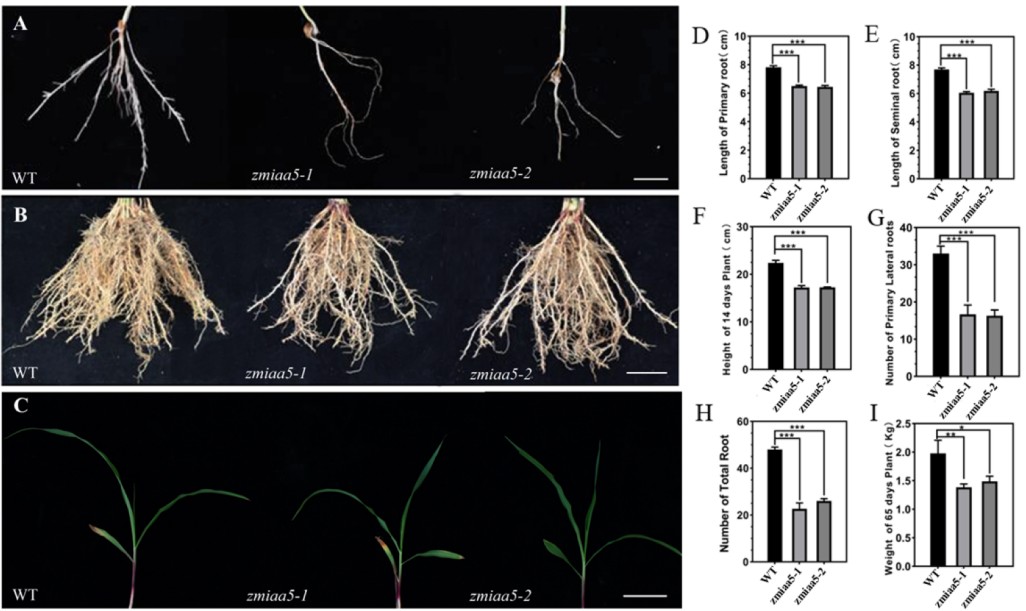

**Figure 4  Phenotypic analysis and statistics of related traits in maize mutant *zmiaa5*.** (A) The root traits of mutants and wild-type seedlings were compared, and the experimental materials were all taken from maize seedlings cultivated for 14 days. (B) The root traits of the mutant and the wild type were compared at the mature stage, and the experimental materials were all taken from maize grown in the field for 65 days. (C) The stalk characteristics of the mutant and the wild type were compared at the seedling stage. The experimental materials were all taken from maize cultivated for 14 days. (D–I) Statistics for maize-related traits at the seedling and mature stages. The statistical data are from three biological replicates and are expressed as the means ± standard deviation, an asterisk (*) indicates a significant difference in results, two asterisks (**) indicates a very significant difference in results. three asterisks (***) indicates extremely significant differences in results.

(Fig. S3A). Immediately, we sent the constructed vector to VIMI Biotechnology Company for transformation and ultimately obtained nine transgenic lines (*Wang & Lambers, 2019*).

We first used the transgenic PAT/bar speed test strip to identify transgenic maize positively and obtained positive results for nine maize lines (Fig. S3A). We further adopted the qRT-PCR experimental method. The expression of the *ZmIAA5* gene in transgenic plants was detected to ensure the accuracy of the experimental results (*Lv et al., 2020*). The results showed that the *ZmIAA5* gene was overexpressed in all nine maize lines (Fig. S3B). We selected the two lines with the highest expression levels, *ZmIAA5*[OE-1] and *ZmIAA5*[OE-2], for phenotypic identification experiments (Fig. S3C).

We found that maize overexpressing *ZmIAA5* had a longer main root length and more lateral roots than wild-type maize (Figs. 5A, 5B and 5C). We performed biological measurements of the taproot length, seed root length, lateral root number, total root number, stem height, and total root dry weight for wild-type and overexpressing maize grown for 14 days in the greenhouse and for 65 days in the field. Following a statistical analysis of data, in all the items under the above statistics, the mutant maize data were lower than the wild-type maize data (Figs. 5D–5I). The results further indicate that the *ZmIAA5* gene is involved in regulating the growth and development of maize roots.

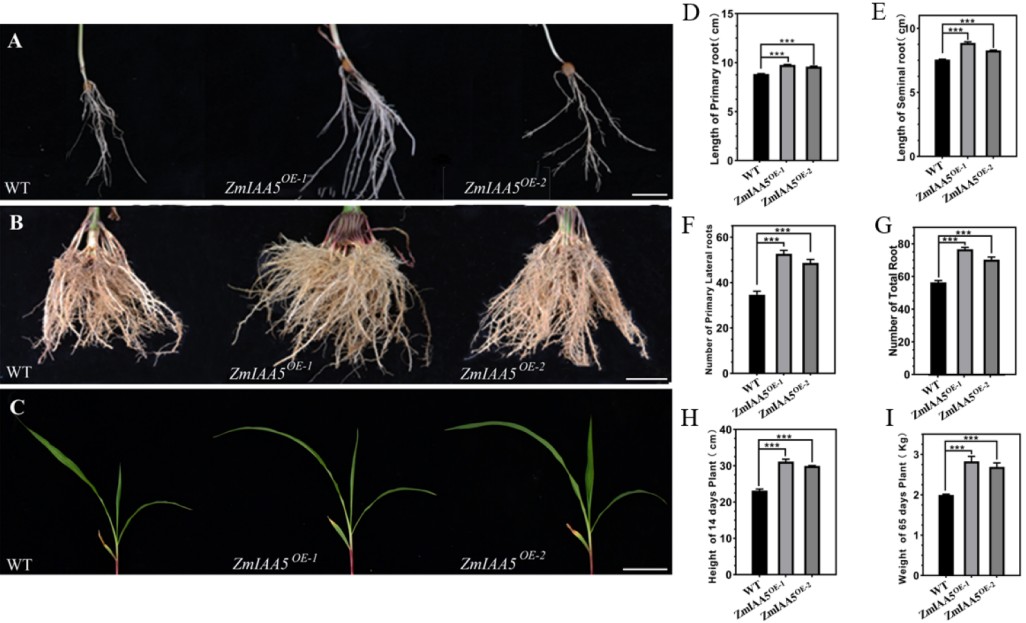

**Figure 5** **Phenotypic analysis and statistics of maize-related traits overexpressing *ZmIAA5*.** (A) For the comparison of root traits between overexpression and wild-type maize seedlings, the experimental materials were all taken from maize seedlings cultured for 14 days. (B) The root traits of the overexpression and wild type maize were compared at maturity, and the experimental materials were all taken from maize grown in the field for 65 days. (C) The stem traits of overexpressed and wild-type plants were compared at the seedling stage. The experimental materials were all taken from maize cultivated for 14 days. (D–I) Statistics of traits related to seedling and maturity stages of overexpressed and wild-type maize. The statistical data are from 3 biological replicates and are expressed as the means ± standard deviation, an asterisk (*) indicates a significant difference in results, asterisk (**) indicates a very significant difference in results. three asterisks (***)* indicates extremely significant differences in results.

### ZmTCP15/16/17 bind to the promoter of *ZmIAA5*

Existing research shows that the *AUX/IAA* gene can be regulated by many genes in plants as an auxin-responsive gene (*Santos et al., 2017*). Among the many reported regulatory genes, the *TCP* family genes are one of the most important types of genes that regulate the *AUX/IAA* family (*Ha & Tran, 2014*). Research on Arabidopsis *TCP* family genes found that the Arabidopsis thaliana class I TCP protein can bind to the GTGGGCCNN sequence on the promoter to regulate downstream genes (*Ha et al., 2012*).

Existing reports show that there are three TCP family genes in maize species, namely, *ZmTCP15, ZmTCP16* and *ZmTCP17* (*Ding et al., 2020*). They have been shown to be involved in auxin signaling responses (*Ding et al., 2020*). Through a sequence analysis of the promoter of the *ZmIAA5* gene, we found that there is a specific binding sequence, GTGGGCCCNN, in the promoter region, and we named this sequence el-IAA5 (Fig. 6A). This specific binding sequence coincides with the binding sequence of the TCP family gene.

To detect whether *ZmTCP15/16/17* binds to the promoter of *ZmIAA5*, Y1H assays were performed. For the Y1H assays, the pGADT7-*ZmTCP15/16/17* and pAbAi-pro*IAA5* vectors were co-transformed into yeast (*Zhang et al., 2022*). pGADT7-Zm*TCP15/16/17*

**A**

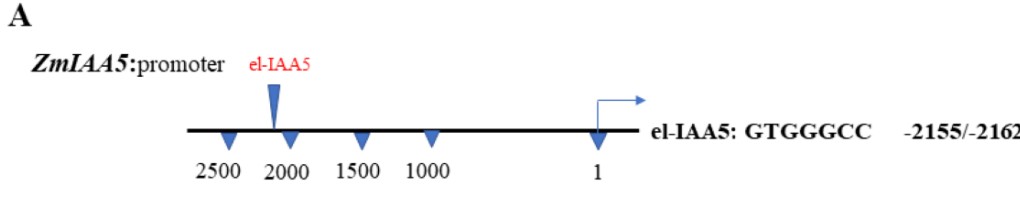

el-IAA5: **ACGATCGTCGATCAGTGGGCC**GCGAGCCGTGGAATTCGGTCC

muel-IAA5: CAGTCTCACCGGCACCTCTTCAATCCCAGTCCCACACAAAG

**B**

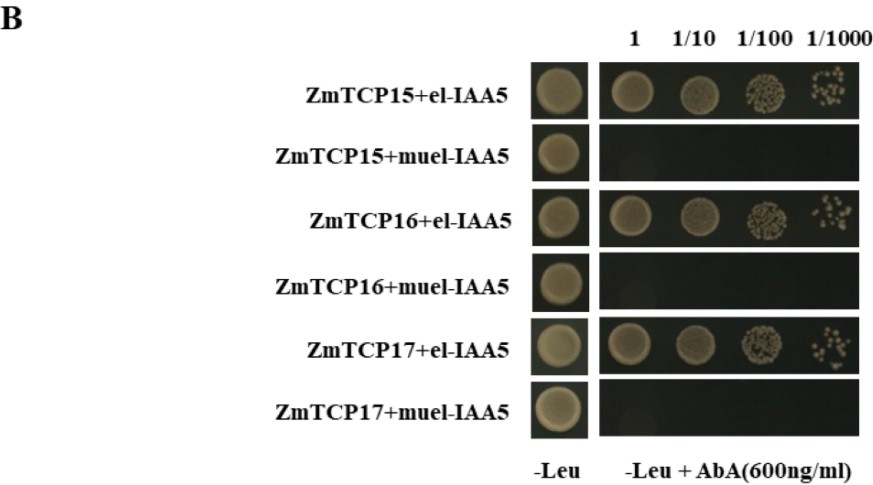

**Figure 6 *ZmTCP15/16/17* binds to the promoter of *ZmIAA5*.** (A) Schematic diagram of the *ZmIAA5* gene promoter. The specific binding sequence in the ZmTCP15/16/17 family is located at −2155 to −2162 of the *ZmIAA5* gene promoter. (B) Y1H assays of the ZmTCP15/16/17 and *ZmIAA5* promoters. Yeast (Y1H) cells containing the indicated plasmids were grown on selective SD-Leu AbA (900 ng/mL) solid medium to test the direct binding of ZmTCP15/16/17 TFs to the promoter of *ZmIAA5*. All the experiments were repeated at least three times with similar results. Error bars indicate ± SD ($n = 3$). Different letters indicate significant differences at $P = 0.032$ according to one-way ANOVA (Tukey's multiple comparison test).

and pAbAi-pro*IAA5* could grow on selective SD-Leu + AbA medium, indicating that ZmTCP15/16/17 interacted with the promoter of ZmIAA5 in yeast (Fig. 6B).

### *ZmIAA5* interacts with *ZmARF5* to regulate maize root growth and development

Studies in Arabidopsis have shown that plant AUX/IAA family proteins interact with ARF family proteins to form dimers, thereby regulating plant life activities (*Li et al., 2018*). Normally, AUX/IAA proteins bind tightly to ARF family proteins. When the plant auxin content increases, the AUX/IAA protein will be degraded by the 26S proteasome, and then the ARF protein will be released (*Sayadi Maazou et al., 2016*). The released ARF protein will activate or inhibit the transcriptional expression of its downstream genes.

It has been reported in the literature that three ARF family proteins in maize may interact with AUX/IAA, namely, *ZmARF5*, *ZmARF7* and *ZmARF25* (*Sayadi Maazou et al., 2016*). To detect whether *ZmARF5/7/25* genes interact with *ZmIAA5*, bimolecular

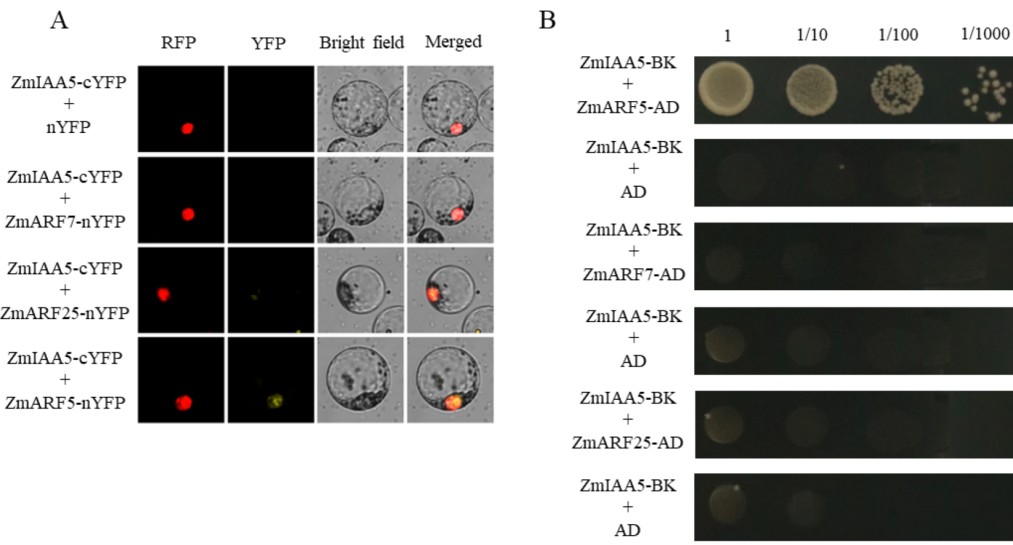

**Figure 7** **Validation of the interaction between ZmIAA5 and ZmARF5/7/25.** (A) Bimolecular fluorescence complementation assays for ZmARF5/7/25 and ZmIAA5. Maize protoplasts showed yellow fluorescence, indicating that ZmIAA5 interacted with ZmARF5. (B) Yeast two-hybrid assays of ZmARF5/7/25 and ZmIAA5. Yeast cells containing the indicated plasmids were grown on selective SD-Leu AbA (900 ng/mL) solid medium to test the interaction of ZmIAA5 with ZmARF5/7/25. All the experiments were repeated at least three times and yielded similar results. Error bars indicate ± SD ($n = 3$). Different letters indicate significant differences at $P = 0.041$ according to one-way ANOVA (Tukey's multiple comparison test).

fluorescence complementation and yeast two-hybrid experiments were performed (*Sayadi Maazou et al., 2016*). For the bimolecular fluorescence complementation, *ZmIAA5*-cYFP and *ZmARF5/7/25*-nYFP vectors were co-transformed into maize protoplasts. A yellow fluorescent signal was detected in the nucleus of the *ZmIAA5*-cYFP and *ZmARF5*-nYFP combination, indicating that ZmIAA5 can interact with ZmARF5; however, ZmIAA5 cannot interact with ZmARF7 and ZmARF25 (Fig. 7A). For the yeast two-hybrid experiments, the ZmIAA-BK and *ZmARF5/7/25*-AD vectors were co-transformed into yeast (*Coelho et al., 2010*). *ZmIAA*-BK and *ZmARF5*-AD could grow on selective SD-Leu + AbA medium, indicating that *ZmIAA5* interacted with *ZmARF5* (Fig. 7B). Additionally, we checked whether the *ZmIAA5* gene has self-activation, and the experimental results showed that it has no self-activation (Fig. S4A).

## DISCUSSION

Maize is one of the most important food crops for human consumption, and molecular research on its growth and development should be given more attention (*Ha et al., 2012*). Previous studies on maize have focused on changes in climate thresholds, nutrient elements, ionic types and hormones in soil (*Ha et al., 2012*; *Ding et al., 2020*; *Li et al., 2018*; *Sayadi Maazou et al., 2016*). Given that maize is a common staple food, fuel, and feed, its yields have continued to increase. However, major regions worldwide rarely exceed 70% of their yield potential (*Lobell, Cassman & Field, 2009*). People are increasingly aware of the

importance of the growth and development of the root system to the health and yield of corn. Research at the molecular level will help people better understand the function of corn genes and make better use of desirable traits (*Aguirre-von Wobeser et al., 2018*).

Roots interact with the highly heterogeneous soil environment and strive against abiotic and biotic stresses to acquire water and nutrients (*Vanhees et al., 2022*). Phytohormones are a class of small but efficient molecules involved in many physiological processes of plant development (*Mehmood et al., 2018*). Indole-3-acetic acid (IAA) is considered to be a biological IAA form of auxin. It has been strongly regulated in plants in several ways, including the biosynthesis, oxidation and hydrolysis of IAA as well as the binding of IAA to macromolecules such as carbohydrates and amino acids (*Perotto, 2011*). At an early stage of auxin signaling transduction, Aux/IAA families are responsive to auxin stimulation (*Wang et al., 2010*). *Aux/IAA* is vital to diverse cellular and developmental processes, including embryogenesis, lateral root initiation, leaf expansion, and fruit development (*Sideris & Young, 1954*; *Kazan & Manners, 2009*).

In this article, we re-identified the maize *Aux/IAA* family genes, including a total of 40 genes, by using bioinformatics methods (Fig. 1A). Sequence alignment and motif alignment analysis showed that maize Aux/IAA proteins primarily shared four conserved domains (Fig. 1B). In accordance with the differences in subfamilies, we selected *ZmIAA1-ZmIAA6* for further research. The experimental results showed that these six genes were located in the nucleus (Fig. 3A), indicating that these six genes played a role in the nucleus.

During the phenotyping experiments, we used maize materials of different genetic backgrounds as a control group. On the one hand, for technical reasons, the KN5585 maize line was used to create overexpressed maize material; on the other hand, the B73 maize line was used to create the maize mutant material. When conducting the experiment, we used the maize lines created with the corresponding materials as the control group, and the experimental results were scientific to some extent.

Previous reports have found that *AUX/IAA* can regulate root development in Arabidopsis (*Schmidt & Schikora, 2001*). However, our studies on the function of the *ZmIAA5* gene in overexpressed and mutant maize indicated that the *ZmIAA5* gene is involved in regulating maize root growth and development. Maize is a drought-tolerant plant, and its growth and development will be under stress in an environment with more water. Many previous reports have used liquid culture for maize cultivation to make it easier to observe maize root tissue. In subsequent experiments, we also observed the root phenotype of maize planted in soil, which further verified our conclusions. Further analysis and experimental study of the *ZmIAA5* promoter revealed upstream genes that may regulate the *ZmIAA5* gene. In addition, we also experimentally verified the target gene *ZmARF5*, which may interact with the *ZmIAA5* gene (Figs. 6 and 7). We preliminarily speculated that there may be a pathway regulating root growth and development in maize. This study provides reliable evidence for further understanding how the *AUX/IAA* family genes are involved in regulating plant growth and development. However, further research is needed on how the interaction between ZmIAA5 and ZmARF5 regulates downstream genes and which genes are regulated to affect root growth and development.

In conclusion, our results show that *ZmIAA5* regulates maize root growth and development by interacting with *ZmARF5* under the specific binding of *ZmTCP15/16/17*.

## ACKNOWLEDGEMENTS

We would like to thank Post-doctoral Yingli Jiang and assistant researcher Xueqiang Su for help.

### Funding

This work was supported by the National Natural Science Foundation of China (U21A20235 to BJ.C). The project name is "Analysis and utilization of key genes function and regulatory mechanism of maize-AM fungal symbiosis to improve nitrogen and phosphorus uptake". The funders had no role in study design, data collection and analysis, decision to publish, or preparation of the manuscript.

### Grant Disclosures

The following grant information was disclosed by the authors:
National Natural Science Foundation of China: U21A20235 to BJ.C.
"Analysis and utilization of key genes function and regulatory mechanism of maize-AM fungal symbiosis to improve nitrogen and phosphorus uptake".

### Competing Interests

The authors declare there are no competing interests.

### Author Contributions

- Feiyang Yang conceived and designed the experiments, performed the experiments, analyzed the data, prepared figures and/or tables, authored or reviewed drafts of the article, and approved the final draft.
- Yutian Shi performed the experiments, analyzed the data, authored or reviewed drafts of the article, and approved the final draft.
- Manli Zhao performed the experiments, prepared figures and/or tables, and approved the final draft.
- Beijiu Cheng conceived and designed the experiments, authored or reviewed drafts of the article, and approved the final draft.
- Xiaoyu Li conceived and designed the experiments, authored or reviewed drafts of the article, and approved the final draft.

### Data Availability

The raw measurements are available in the Supplementary Files.

### Supplemental Information

Supplemental information for this article can be found online at http://dx.doi.org/10.7717/peerj.13710#supplemental-information.

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
