# Peer review of "ZmIAA5 regulates maize root growth and development by interacting with ZmARF5 under the specific binding of ZmTCP15/16/17"

_PeerJ, doi:10.7717/peerj.13710_

## Round 0.1 · original submission · Major Revisions

This manuscript is somewhat innovative but requires major revisions.

Reviewer 1 ·

Basic reporting

In this study, the authors comprehensively studied the ZmIAA5 by phenotypic analysis of overexpression/mutant lines and identification of its regulatory gene as well as interacting proteins. The study shows some applicability and novelty. This manuscript has been prepared well. While there are some minor problems need to be revised before it can be acceptable for PeerJ.
1. Introduction is insufficient and needs to be further enriched.
2. The figures resolution is too low to be clear.
3. There are some inaccurate and confusing English usages in the manuscript. The authors should revise their paper carefully.

Experimental design

No question

Validity of the findings

No question

Reviewer 2 ·

Basic reporting

The authors of this manuscript investigated ZmIAA5 regulates maize root growth and development by interacting with ZmARF5 under the specific binding of ZmTCP15/16/17.Through the study of the overexpression of the ZmIAA5 gene and the phenotype of mutants in maize, combined with a series of molecular experiments, the regulatory relationship between the upstream and downstream of the gene was elucidated. The article is clear in thought and detailed in the data but I have several concerns on the presented data as described below.

Experimental design

1. The background part of the article abstract should be enriched.
2. The conclusion part of the article abstract should be described in more detail.
3. The manuscript would benefit from some improvements in English writing. The text includes a few typing errors and some minor grammar mistakes.
4. The format of gene names in the article needs to be uniform, some in italics, some without italics.
5. Some of the figures in the manuscript are not clear, and it is recommended to replace the clearer figures to facilitate the reader to clearly obtain the results of the experiment.
6. What is the main formula of liquid medium for cultivating corn? It should be reflected in the materials and methods section.

Validity of the findings

7. The growth of maize tends to be more in arid environments, and whether culturing maize in liquid medium will cause stress to maize, the authors need to explain in the article.
8. The data statistics on the root phenotype of maize at 65 days of growth are carried out in the article, and the conditions and processes of maize growth should be explained in the article.
9. The conclusion section of the article can be added to a regulatory pathway figure to more intuitively represent the conclusion.
10. The drawing in Figure 5C is not standardized, and the added text obscures the picture.
11. The red ellipse callout in Figure 2S should be modified and should not extend beyond the main body.
12. The icon in Figure S3B should be readjusted to make the picture more aesthetically pleasing.

·

Basic reporting

no comment

Experimental design

no comment

Validity of the findings

no comment

Additional comments

The authors report the function of the ZmIAA5 gene and its involvement in new pathways that regulate maize root growth and development. This article is well written and contains useful and interesting data, the data presented supports the conclusions reached in the manuscript. However, I have a few issues that the authors should address before publication.
1.Pay attention to the problem of language expression, and the authors need to find a fluent English speaker to modify it.
2.The issue of writing format should be taken seriously. For example, when gene names appear in the article, some use italics, some not, the format should be unified.
3.What does EMS mean? The full name should be used when writing for the first time.
4.The article mentions many gene names, and each gene should be marked with its original gene number when it first appears.
5.Figure 1, I suggest using a clearer picture to facilitate the reader's observation.
6.Figure 4 and Figure 5 needs to indicate repeat number of the experiment.
7.Typically, the same control maize lines are required to be used for the experiments, but the control maize lines selected in Figures 4 and 5 are different (B73 and KN5585) and should be explained by the authors.
8.In Figures 5G and 5E, attention should be paid to the distance between the ordinate labels and the histogram
9.There is a typo in Figure 6B, "elIAA5" should be changed to "el-IAA5".
10.The text description in the article and the mark in the picture should be unified. For example, in the Supplementary Material part of the article, a, b, c... are used, while in the figure, A, B, C...

Reviewer 4 ·

Basic reporting

In this study, the authors comprehensively studied the ZmIAA5 by phenotypic analysis of overexpression/mutant lines and identification of its regulatory gene as well as interacting proteins. This manuscript is somewhat innovative, but requires major revisions to be accepted by PeerJ.

Experimental design

No comment

Validity of the findings

No comment

Additional comments

1. There are some inaccurate and confusing English usages in the manuscript. The authors should revise their paper carefully.
2. When a protein or gene appears for the first time in the manuscript, please use the full text.
3. Line36,37. The word “Maize” repeated.
4. Line188,189,190. Gene names need to be italicized.
5. The figures resolution is too low to be clear.
6. Authors need to unify usage of the word “corn” or “maize”.
7. Gene names need to be italicized, but protein names do not, and authors should check the manuscript carefully.
8. In the results of the phenotypic experiments, why not use maize material from the same background as a control? The author should explain reason in the manuscript.
9. In the phenotypic results, the authors should select maize with the same growth period to observe the plant height phenotype.
10. Line 490. Please take care to check the writing of the references, remove the question mark.

---

## Round 0.2 · accepted · Accept

Congratulations!
Thanks for your contribution to this journal.

·

Basic reporting

no comment

Experimental design

no comment

Validity of the findings

no comment

Additional comments

the authors revised the manuscript carefully, the article answered all the questions in my comments.